# CROSS-BATCH GRADIENT CONSISTENCY FOR ADAPTIVE LOSS BALANCING IN KNOWLEDGE DISTILLATION

## ABSTRACT

Knowledge distillation (KD) is a widely used approach for compressing large neural networks into compact student models by combining supervised learning with teacher-guided alignment. While recent studies have attempted to improve KD through adaptive weighting between the supervised and distillation objectives, most existing methods determine weights solely from gradients computed on a single mini-batch. This batch-local perspective neglects the crucial requirement that student updates should generalize across unseen data, often resulting in gradient conflicts, unstable training dynamics, and suboptimal performance. In this work, we introduce a cross-batch dynamic weighting framework for KD that explicitly incorporates generalization signals beyond the current batch. At each iteration, we leverage an auxiliary batch as a proxy for unseen data, compute its supervised gradient as a reference, and solve a lightweight quadratic program to adaptively select weights that align the combined update direction with this reference. To further stabilize optimization, we normalize task gradients and introduce a scaling mechanism that balances their magnitudes while maintaining computational efficiency. Extensive experiments on standard benchmarks demonstrate that our approach consistently outperforms fixed-weight and batch-local adaptive baselines, leading to more stable optimization and superior student performance. These results highlight the importance of cross-batch consistency in KD and establish our method as a principled and effective strategy for dynamic loss balancing.

## 1 INTRODUCTION

Knowledge distillation (KD) [15] has emerged as a powerful paradigm for compressing large teacher neural networks into lightweight student models while retaining competitive performance. In an era dominated by ever-growing large-scale models like large language models and vision transformers, KD remains a crucial and widely adopted technique for deploying these complex models in resource-constrained environments. By leveraging both the ground-truth labels and the guidance of a teacher model, KD provides a flexible mechanism for improving generalization and efficiency in student training.

KD typically trains a student model by jointly optimizing two objectives: the supervised cross-entropy (CE) loss and the Kullback–Leibler (KL) divergence loss. As there are two loss terms, setting appropriate loss weights is crucial for KD and conventional methods usually search static loss weights to combine two terms. However, recent studies [50, 46, 30, 12, 44, 26] have shown that using fixed loss weights fails to account for the varying importance of the tasks across training iterations, leading to either under- or over-emphasis of the distillation signal relative to the supervised signal, and thus is often suboptimal. To this end, recently, researchers have shifted their attention to designing dynamic weighting strategies. In [50], Yu et al. introduced instance-wise weighting based on the discrepancy between the teacher's predicted probability and the ground-truth label, while [46] leveraged adaptive weighting by focusing on the teacher's key predictive regions in object detection. [30] developed an adaptive loss weighting strategy for semantic segmentation by gradually reducing the teacher's influence, and [12] dynamically adjusts the KD loss weight by assigning a per-instance weight based on a sample's difficulty, which prioritizes distilling knowledge from easier samples with a lower loss weight, while increasing the weight for harder samples as training progresses.

More recently, [44] estimate the influence of a sample on unseen data by by multiplying the inverse Hessian of the training loss with the sample's loss gradient for dynamic loss weighting. [47] prioritizes the separation of positive and negative documents by down-weighting the loss for well-ranked positives and poorly-ranked negatives, while emphasizing the alignment with the teacher for documents that the student model struggles with. [26] adjusts KL loss weights based on the discrepancy between the student and teacher's probability distributions, shifting the focus towards hard-label supervision when the student's output differs significantly from the teacher's.

While prior dynamic weighting methods have shown improvements by adjusting loss weights at the instance or mini-batch level, they generally rely only on signals from the current batch. Such batch-local strategies implicitly assume that optimizing the weighted loss within one batch automatically leads to better generalization, but this assumption is not guaranteed. In fact, the two objectives in KD—the CE loss tied to supervised accuracy and the KL loss serving as auxiliary teacher alignment—often produce gradients that are not aligned, causing destructive interference and unstable updates [28, 49]. Importantly, what ultimately matters is not just minimizing the weighted loss on current samples, but improving the primary CE objective on unseen data. Existing methods, however, lack an explicit mechanism to ensure that their weighting improves CE generalization, and thus risk either over-emphasizing or under-utilizing the teacher signal in ways that do not benefit the student's predictive accuracy.

To overcome these limitations, we propose a new dynamic weighting framework that explicitly optimizes for generalization of the CE task on unseen samples, rather than only reconciling CE and KL within a single batch. Our key idea is to leverage an auxiliary batch as a proxy for unseen data and select weights such that the weighted update direction from the current batch is consistent with the CE gradient on the auxiliary batch. This design ensures that the estimated loss weights directly promote improvement of the primary supervised task, while still incorporating the auxiliary distillation signal. Unlike multi-task learning (MTL) methods [19, 11], which aim to balance multiple tasks of equal importance, our formulation explicitly prioritizes the CE objective and treats KL as a secondary regularizer, reflecting the asymmetric roles of the two losses in KD. Furthermore, since CE and KL gradients often differ in scale due to their heterogeneous objectives, we integrate gradient normalization and a scaling mechanism [28, 11] to prevent one task from dominating merely because of magnitude imbalance. This ensures that the dynamic weighting reflects genuine trade-offs rather than artifacts of gradient norms. Together, these components yield a principled and stable optimization scheme, enabling more consistent knowledge transfer and improved generalization performance of the student model.

The main contributions of this work are summarized as follows:

- We identify the overlooked limitation of existing KD approaches that rely on batch-local weighting, which often leads to unstable optimization due to gradient conflicts. To address this issue, we propose a novel cross-batch dynamic weighting framework that leverages auxiliary gradients from an additional batch to adaptively guide the selection of task weights at every iteration.

- We develop an efficient optimization procedure with gradient normalization and scaling, ensuring stable updates while incurring negligible computational overhead. This design maintains task balance and effectively integrates CE and KL objectives.

- Extensive experiments demonstrate that our method consistently improves student model performance over fixed-weight and batch-local adaptive baselines, validating the effectiveness of our cross-batch perspective.

## 2 METHODOLOGY

### 2.1 KNOWLEDGE DISTILLATION WITH DYNAMIC WEIGHTING

In a typical knowledge distillation (KD) framework, the training of the student model $S$ involves two complementary objectives. The first is the cross-entropy (CE) loss, which enforces prediction accuracy with respect to the ground-truth labels. Given a batch $B = (x_i, y_i)_{i=1}^{m}$ of inputs $x_i$ and

labels $y_i$, the CE objective is

$$\mathcal{L}_{\text{ce}}(B;S) = -\frac{1}{m}\sum_{i=1}^{m}\sum_{c=1}^{C}\mathbf{1}[y_i = c]\log p_S(c|x_i), \tag{1}$$

where $p_S(c \mid x)$ denotes the predictive distribution of $S$ over the $C$ classes. The second component is the distillation loss, which typically aligns the student with the teacher $T$ at a distributional or representational level. We denote this generically as a Kullback–Leibler divergence,

$$\mathcal{L}_{\text{kl}}(B;S,T) = \frac{1}{m}\sum_{i=1}^{m}D_{\text{KL}}(q_T(z|x_i)\|q_S(z|x_i)), \tag{2}$$

where $q_T(\cdot \mid x)$ and $q_S(\cdot \mid x)$ represent teacher and student distributions over some latent variable $z$. The choice of $z$ is flexible: it may correspond to softened class probabilities, intermediate hidden features, or attention maps, depending on the particular KD variant employed. Combining the two terms, the overall training objective in conventional KD is

$$\mathcal{L}(B;S) = w_{\text{ce}}\mathcal{L}_{\text{ce}}(B;S) + w_{\text{kl}}\mathcal{L}_{\text{kl}}(B;S,T), \tag{3}$$

with nonnegative coefficients $w_{\text{ce}}, w_{\text{kl}}$ that are typically fixed beforehand and remain constant across all training iterations.

However, such a fixed weighting scheme lacks adaptivity and may lead to suboptimal updates. Since $g_{\text{ce}}, g_{\text{kd}}$ (the gradients of $\mathcal{L}_{\text{ce}}$ and $\mathcal{L}_{\text{kl}}$) are computed from stochastic mini-batches, their directions often exhibit variability and even conflict across iterations. A constant convex combination $w_{\text{ce}}g_{\text{ce}} + w_{\text{kl}}g_{\text{kl}}$ cannot flexibly adapt to these fluctuations, which in turn deteriorates optimization dynamics and slows down convergence. In particular, a weight setting that is beneficial on one batch can bias the training signal on another batch where the relative gradient scales and directions differ.

To address this issue, we propose to dynamically determine the task weights at every training step. Specifically, in addition to the current batch $B_1$ used to compute gradients for both $\mathcal{L}_{\text{ce}}$ and $\mathcal{L}_{\text{kl}}$, we draw an auxiliary batch $B_2$ from the same dataset (with independent shuffling). The core idea is to evaluate how well a candidate update on $B_1$ generalizes to $B_2$ under the CE objective, and to adaptively select weights $w = [w_{\text{ce}}, w_{\text{kl}}]$ that minimize the inconsistency between the combined update direction and the CE gradient on $B_2$. This construction provides a mechanism for aligning the weighted task gradient with the direction that is most consistent across different samples, thereby mitigating gradient conflicts and enabling more effective student optimization.

## 2.2 Weight Searching for Improvements on Unseen Data

We begin by introducing the weighted gradient of the student model as

$$g_w = w_{\text{ce}}g_{\text{ce}} + w_{\text{kl}}g_{\text{kl}}, \tag{4}$$

where $g_{\text{ce}}$ and $g_{\text{kl}}$ denote the task-specific gradients of the CE loss and the KL-based distillation loss, respectively. The coefficients $w_{\text{ce}}$ and $w_{\text{kl}}$ regulate the relative contribution of each task which are adaptively optimized at every iteration. The central question is how to select $w = (w_{\text{ce}}, w_{\text{kl}})$ such that the resulting update direction is both effective on the current batch and consistent with unseen data.

To address this, we introduce an auxiliary signal from an unseen batch. Specifically, we freeze the student parameters and compute a reference gradient from the CE loss on a different batch, denoted by $\mathcal{G}_{\text{ce}}$. This auxiliary gradient serves as a proxy for the generalization behavior of the model, since it reflects how the student would respond to samples not directly involved in the current update. Our objective is to prevent the weighted update from overly aligning with this auxiliary signal, thereby reducing gradient conflicts between batches. Formally, we minimize:

$$\min_{w \in \mathcal{W}}(g_w^{\top}\mathcal{G}_{\text{ce}})^2, \quad \mathcal{W} = \{w_{\text{ce}}, w_{\text{kl}} \geq 0, w_{\text{ce}} + w_{\text{kl}} \leq 1\}. \tag{5}$$

This problem can be compactly written as a quadratic program of the form

$$\min_{w \in \mathcal{W}}(u^{\top}w)^2, \quad u = [g_{\text{ce}}^{\top}\mathcal{G}_{\text{ce}}, g_{\text{kl}}^{\top}\mathcal{G}_{\text{ce}}]^{\top}. \tag{6}$$

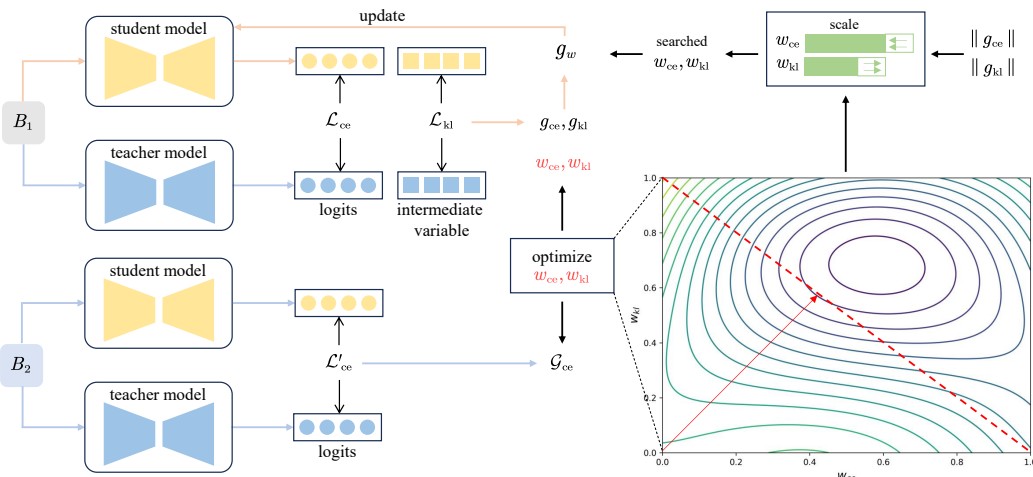

Figure 1: Overall methodology of the proposed dynamic weighting framework. Two batches $B_1$ and $B_2$ are sampled to compute task gradients $(g_{ce}, g_{kl})$ and the auxiliary CE gradient $\mathcal{G}_{ce}$. We adaptively search weights $(w_{ce}, w_{kl})$ to minimize cross-batch inconsistency, followed by gradient normalization and a scaling factor $\lambda$ to stabilize the final update.

The matrix $uu^\top$ is positive semidefinite with rank one, making the optimization convex and computationally lightweight.

In our design, gradients from other mini-batches only involve the CE loss. This choice is motivated by the role of CE and KL in distillation: CE constitutes the primary objective directly tied to task performance, while KL merely serves as an auxiliary alignment signal. Incorporating KL across batches would risk over-emphasizing the auxiliary task and weakening the focus on the supervised objective. By restricting cross-batch aggregation to CE, we ensure that the optimization remains centered on the main learning target, with KL acting only as a within-batch regularizer.

By construction, this formulation adaptively selects task weights that minimize destructive interference between CE and KL updates, ensuring that the student's gradient direction remains stable across batches. In contrast to fixed weighting schemes, our approach dynamically reconciles the two tasks in a data-dependent manner, yielding more consistent and generalizable knowledge transfer.

## 2.3 Gradients Balance and Stabilization

Once the optimal coefficients $w = (w_{ce}, w_{kl})$ are obtained from the quadratic program in Eq. equation 6, we construct the weighted task gradients as

$$g'_{ce} = w_{ce}g_{ce}, \quad g'_{kl} = w_{kl}g_{kl}, \tag{7}$$

**Gradient Balancing.** Due to the inherent imbalance in task gradient magnitudes, a direct combination often leads to instability where one task dominates the update solely because of its scale. This effect is particularly evident in KD, as CE and KL losses are defined on heterogeneous objectives and thus yield gradients with substantially different magnitudes [19, 11, 28]. Without correction, the task with the larger gradient norm would overshadow the other, biasing the optimization regardless of the adaptive weight search. To address this, we normalize the task contributions by rescaling the weights as

$$\tilde{w}_{ce} = w_{ce} \cdot \frac{\|g'_{ce}\|_2 + \|g'_{kl}\|_2}{\|g'_{ce}\|_2}, \qquad \tilde{w}_{kl} = w_{kl} \cdot \frac{\|g'_{ce}\|_2 + \|g'_{kl}\|_2}{\|g'_{kl}\|_2}. \tag{8}$$

This rescales $w_{ce}$ and $w_{kl}$ proportionally to the relative task gradient magnitudes, so that the effective contributions $g'_{ce}$ and $g'_{kl}$ are comparable in norm. This step prevents raw scale discrepancies from dictating the update direction and ensures that the adaptive weighting reflects genuine task trade-offs rather than artifacts of heterogeneous loss formulations. Such normalization has been shown to mitigate gradient dominance and improve stability across objectives [19, 11, 28].

We then scale the loss weights by the weight of CE such that the CE weight is 1 and KL weight will adjust the contribution of the KL loss:

$$w_{\text{ce}}^{\text{final}} = 1, \qquad w_{\text{kl}}^{\text{final}} = \frac{\tilde{w}_{\text{kl}}}{\tilde{w}_{\text{ce}}}. \tag{9}$$

We observed that CE constitutes the primary supervised objective directly governing the student's predictive accuracy, while KL serves as an auxiliary alignment signal guiding the student to mimic the teacher, as confirmed by experiments in Section 3.4. By fixing $w_{\text{ce}} = 1$, we guarantee that the main learning signal remains dominant, while the proportional scaling preserves the relative contribution of the auxiliary KL task identified by dynamic weighting. This strategy aligns with common practices in multi-task learning, where primary task gradients are maintained at a stable magnitude while secondary tasks are adaptively scaled [11, 19, 28]. Empirically, this adjustment improves convergence stability and ensures that the student benefits from the teacher's guidance without compromising task performance.

**Gradient Stabilization.** Beyond this final weighting, we further stabilize the optimization by introducing a scaling factor $\lambda$ inspired by CAGrad [28]. Let $g_0 = \frac{1}{2}(g_{\text{ce}} + g_{\text{kl}})$ denote the unbiased mean gradient across tasks, and let $g_w = w_{\text{ce}}^{\text{final}} g_{\text{ce}} + w_{\text{kl}}^{\text{final}} g_{\text{kl}}$ denote the dynamically weighted gradient. We define

$$\lambda = \frac{\alpha \|g_0\|_2}{\|g_w\|_2}, \tag{10}$$

where $\alpha$ is a hyperparameter controlling the relative strength of correction. This formulation ensures that the additional signal from $g_w$ is scaled in proportion to the reference magnitude of $g_0$, preventing excessively large or small updates due to fluctuating weight solutions. The final update rule becomes

$$g_{\text{final}} = g_0 + \lambda g_w, \quad S \leftarrow S - \eta g_{\text{final}}, \tag{11}$$

with $\eta$ the learning rate. Intuitively, Eq. (11) interpolates between the unbiased mean gradient $g_0$ and the adaptively weighted correction $g_w$: when $\lambda$ is small, the update is close to $g_0$, while larger values emphasize the task-adaptive direction. This balancing mechanism ensures that dynamic weighting improves optimization without destabilizing it, yielding updates that are both data-dependent and scale-consistent.

In practice, each iteration requires two forward–backward passes: one on $B_2$ with frozen student parameters to obtain $\mathcal{G}_{\text{ce}}$, and one on $B_1$ with trainable parameters to compute $g_{\text{ce}}$ and $g_{\text{kl}}$. The additional quadratic program and rescaling operations incur negligible computational overhead, while the incorporation of $\lambda$ provides consistent training stability across tasks.

## 3 EXPERIMENTS

**Datasets.** We conduct experiments on CIFAR-100 [20]. CIFAR-100 is a widely used image classification dataset consisting of 100 classes, with each image sized 32×32 pixels. It contains 50,000 training images and 10,000 test images.

**Baselines.** We evaluate the effect on our improved CAGrad method for conventional knowledge distillation approaches, including KD [15], RKD [35], AT [51], KDSVD [22], PKT [36]. These methods are implemented using the mdistiller [54, 55] repository. For each experimental setting, we run the repository's distillation training code with adjusted parameters to obtain a trained student model, whose classification accuracy serves as the baselines.

**Implementation Details.** All experiments were conducted on a single NVIDIA GeForce RTX 3090 GPU. We strictly follow the training and testing protocol, and hyperparameters in the prior work [18, 10]. To ensure a fair comparison, we kept all shared training parameters identical between the baseline and our proposed method. Specifically, for all distillation methods, we used a batch size of 64, trained for 240 epochs, initialized the learning rate at 0.05, applied learning rate decay at epochs 150, 180, and 210 with a decay factor of 0.1, and used SGD as the optimizer. All results are reported by taking average over 5 trials.

Table 1: Comparisons of student model performance between conventional KD methods and our proposed approach.

| Teacher
Student | resnet101
resnet50 | resnet56
resnet20 | vgg13
vgg8 | resnet101
resnet34 | resnet110
resnet32x4 | resnet110
resnet56 | wrn-40-2
wrn-40-1 | Avg % |
|---|---|---|---|---|---|---|---|---|
| KD | 80.21 | 70.62 | 72.90 | 80.57 | 78.86 | 75.14 | 73.65 | 75.99 |
| KD + ours | **80.51** | **71.59** | **73.46** | **81.22** | **79.31** | **75.72** | **74.83** | **76.66** |
| RKD | 80.18 | 70.12 | 71.45 | 79.56 | 79.36 | 74.38 | 72.02 | 75.30 |
| RKD + ours | **80.83** | **70.93** | **72.09** | **80.40** | **80.05** | **75.09** | **72.76** | **76.02** |
| AT | 79.93 | 69.70 | 71.54 | 79.85 | 79.18 | 73.92 | 73.02 | 75.31 |
| AT + ours | **80.23** | **70.74** | **72.64** | **80.61** | **79.53** | **74.62** | **74.01** | **76.05** |
| KDSVD | 78.73 | 70.11 | 70.53 | 79.42 | 79.62 | 73.06 | 71.83 | 74.76 |
| KDSVD + ours | **79.70** | **71.07** | **71.39** | **80.13** | **80.26** | **73.52** | **72.58** | **75.52** |
| PKT | 80.32 | 70.43 | 73.19 | 80.22 | 80.15 | 74.70 | 73.73 | 76.11 |
| PKT + ours | **80.95** | **71.20** | **73.77** | **80.94** | **80.80** | **75.39** | **74.09** | **76.73** |

## 3.1 RESULTS OF OUR PROPOSED APPROACH

We first evaluate our proposed dynamic weighting strategy by integrating it into five representative knowledge distillation approaches: KD [15], RKD [35], AT [51], KDSVD [22], PKT [36]. To ensure generality, experiments are conducted on seven widely adopted teacher–student pairs, including resnet, vgg, and wrn families, ranging from large-capacity teachers such as resnet101 and resnet110 to compact students such as resnet20 and wrn-40-1.

Table 1 reports the top-1 accuracy of student models under conventional distillation and under our method. For each baseline, the best result between the vanilla and "+ ours" variant is highlighted in bold. The last column summarizes the average improvement across all pairs. Overall, our method consistently improves student performance across different distillation techniques and architectures. A closer inspection reveals that the gains are particularly pronounced on challenging small-student settings (e.g., ResNet-56→ResNet-20 and WRN-40-2→WRN-40-1), where fixed weighting often struggles to balance CE and KL objectives. By contrast, our method dynamically adapts the task contributions at each iteration, leading to more effective knowledge transfer. These results verify that the proposed dynamic weighting not only generalizes well to diverse distillation paradigms but also provides consistent benefits regardless of the teacher–student gap, thereby highlighting its practical value for real-world deployment.

To further verify the robustness of our method, we additionally conduct experiments on the Tiny-ImageNet dataset, which is more challenging due to its larger scale and higher variability. We evaluate several teacher–student pairs using the same experimental setup. As shown in **??**, our approach again achieves consistent improvements over standard KD across all pairs. These results confirm that our dynamic weighting strategy is not limited to CIFAR benchmarks but also generalizes well to larger and more complex datasets.

## 3.2 KNOWLEDGE DISTILLATION METHODS ARE SENSITIVE TO LOSS WEIGHTS

Knowledge distillation optimizes the model for minimizing the weighted average losses of the primary task and distillation. Here, we conduct experiments by using the standard KD method over various teacher-student pairs: wrn-40-2 and wrn-40-1, resnet101 and resnet50, vgg13 and vgg8. This aims to comprehensively investigate the effect of loss weights on KD with various model sizes and architectures. In additional to this, to ensure generality and eliminate the influence of any specific distillation method, we further experiment with RKD, KDSVD, and PKT using the vgg13–vgg8 teacher-student pair. For all experiments, we train each model with various loss weights by varying $\alpha_{\text{CE}}$ from 0.1 to 0.9 in equal increments, with $\alpha_{\text{KL}} = 1 - \alpha_{\text{CE}}$.

The results are shown in Fig. 2. To present the results more intuitively, we adopt different visualization strategies for the two figures. In the left figure, due to the large variation in distillation performance across settings, we present the accuracy difference value (positive value means higher accuracy) between the model with a loss weight and the one with uniform loss weight ratio (CE vs KL : 0.5 vs 0.5). In the right figure, since different distillation methods use different initial loss weight ratio (e.g., CE vs KL: 1 vs 30000 for PKT and 1 vs 1 for RKD and KDSVD), we use scaling

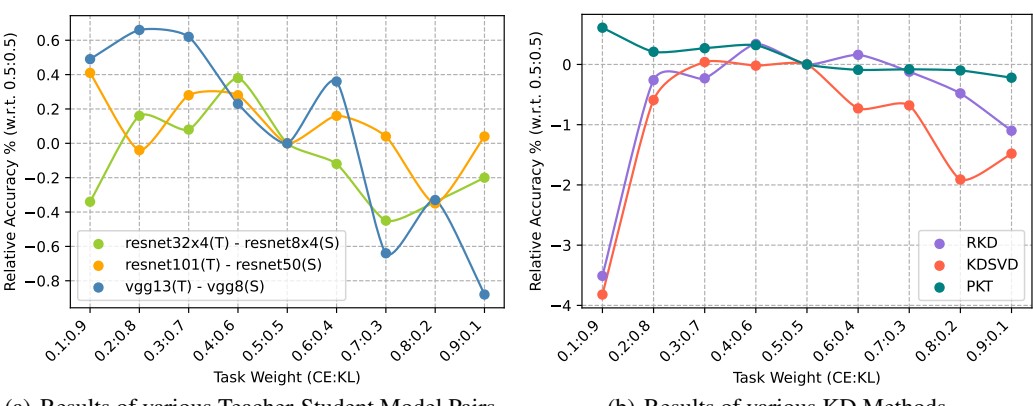

(a) Results of various Teacher-Student Model Pairs.

(b) Results of various KD Methods.

Figure 2: Distillation performance under varying loss weight ratios. (a): Accuracy of student models trained with the KD method across different teacher–student pairs under varying CE/KL loss weight ratios. (b): Accuracy trends for the vgg13-to-vgg8 distillation setting using different distillation methods. Note that the initial loss weight for the RKD and KDSVD methods is 1.0:1.0, while for the PKT method it is 1.0:30000.0. To unify the representation of the x-axis, we use the proportion of task importance. For example, when x is set to 0.5:0.5, it corresponds to a weight ratio of 1.0:1.0 for RKD and KDSVD, and 1.0:30000.0 for PKT; when x is set to 0.3:0.7, the corresponding weight ratios are 0.6:1.4 and 0.6:42000.0, respectively.

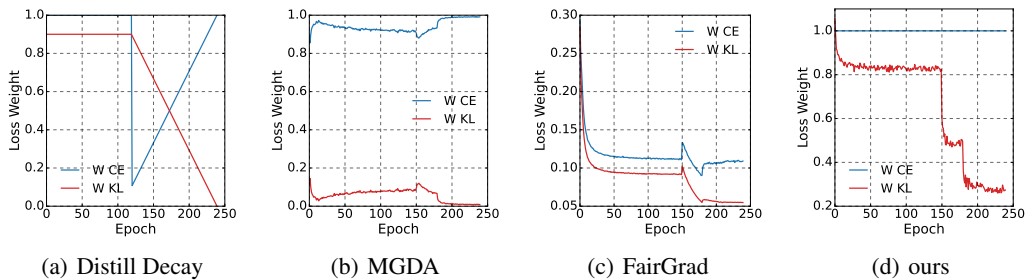

(a) Distill Decay

(b) MGDA

(c) FairGrad

(d) ours

Figure 3: Loss weighting dynamics of different strategies. Our method achieves smoother and more interpretable adjustments compared to existing baselines.

factors on the x-axis to represent how the original weights are adjusted, rather than displaying the absolute weight values. These results show that the commonly used 0.5:0.5 loss weight ratios in distillation is often suboptimal, and alternative ratios can yield better performance. This underscores the limitation of fixed weighting and reinforces the motivation for our approach, which automatically adjusts the loss weights during training, leading us to introduce multi-task learning methods into the distillation task.

## 3.3 COMPARISON WITH DYNAMIC WEIGHTING APPROACHES

To further validate the effectiveness of our method, we draw connections between knowledge distillation and multi-task learning (MTL), where the CE and KL objectives can be viewed as two tasks to be dynamically balanced. Accordingly, we compare our approach against several representative MTL-based weighting strategies: MGDA [39], which finds Pareto-optimal updates across tasks, FairGrad [4], which enforces fairness in gradient contributions, and PCGrad [49], which resolves gradient conflicts by projection. We also include a simple distillation-decay baseline, which gradually reduces the KL loss weight $\alpha$ (initialized as 0.9) to 0 and increases the CE loss weight $\gamma$ (initialized as $1.0 - \alpha = 0.1$) to 1.0 after the midpoint of training, while additionally introducing a KD regularization term with coefficient $\beta$ in later epochs.

Table 2: Comparison of student model performance between our dynamic weighting method and existing task-specific KD dynamic weighting approaches (PGD, AICSD, SnT) on standard image classification benchmarks. Top-1 accuracy (%) is reported.

| Teacher Model | Student Model | original | distill-decay | MGDA | FairGrad | PCGrad | ours |
|---|---|---|---|---|---|---|---|
| resnet101 | resnet50 | _80.21_ | 79.97 | 79.43 | 81.47 | 79.12 | **80.51** |
| resnet56 | resnet20 | 70.62 | 71.54 | **71.77** | 70.00 | 71.54 | _71.59_ |
| wrn-40-2 | wrn-40-1 | 73.65 | 74.02 | _74.60_ | 73.25 | 74.10 | **74.83** |
| resnet32x4 | resnet8x4 | 73.32 | 73.80 | 72.77 | 72.89 | **75.77** | _74.67_ |
| average | | 74.45 | 74.83 | 74.64 | 74.40 | _75.13_ | **75.40** |

For fair evaluation, all methods are implemented under a unified training framework with identical teacher–student pairs and datasets. Hyperparameters are tuned following the recommended settings of each baseline. We report top-1 classification accuracy as the primary evaluation metric. Results are summarized in Table 2, where our method consistently achieves competitive or superior performance compared to existing dynamic weighting strategies, highlighting its robustness and effectiveness in reconciling supervised and distillation objectives. As shown in Figure **??**, Distill Decay relies on a fixed schedule with abrupt shifts, MGDA almost ignores the KL term, and FairGrad suffers from unstable oscillations. In contrast, our method adaptively reduces the KL weight in a smooth manner while keeping CE dominant, leading to stable and interpretable training dynamics.

## 3.4 ABLATION STUDIES

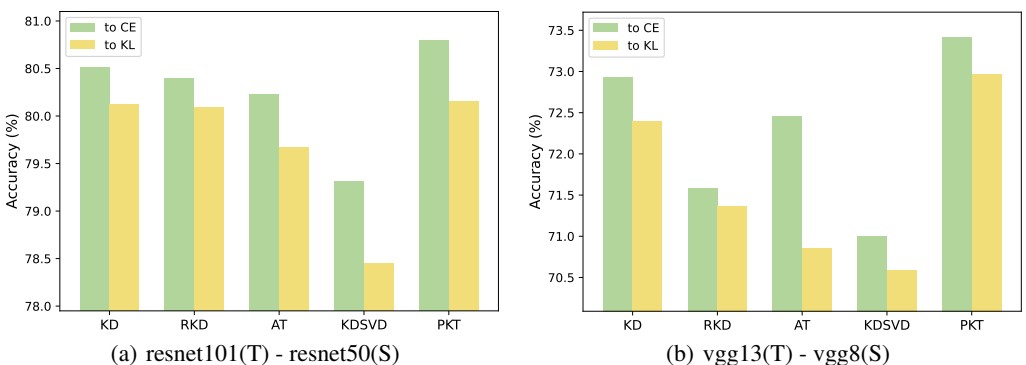

(a) resnet101(T) - resnet50(S)

(b) vgg13(T) - vgg8(S)

Figure 4: Ablation Study on prefering towards primary tasks or distillation tasks. We consider two teacher-student model pairs: (a) resnet101 (teacher) and resnet50 (student), and (b) vgg13 (teacher) and vgg8 (student). It can be seen that, under otherwise identical conditions, setting the primary task as the preference generally yields better results than setting the distillation task as the preference.

**Task preference term in task optimization.** In Section 2.3, we introduce a task preference term that steers the optimization toward gradients more aligned with the primary task. In this section, we explore the effect of this preference by adjusting it to favor either the primary or the distillation task, while keeping all other settings unchanged. We conduct comparative experiments using the KD method across different teacher–student model pairs, as shown in Fig. 4. The results show that preference toward the distillation task does not outperform prefering toward the primary task. This is because the distillation objective is intended to indirectly enhance performance of primary tasks by encouraging the student to mimic the teacher's intermediate outputs and improve generalization. However, the teacher is not always reliable and may produce misleading signals that conflict with the gradient direction of the primary task. As a result, prioritizing the distillation task through the task preference term can harm the student's classification accuracy and lead to unstable training.

## 4 RELATED WORKS

### 4.1 KNOWLEDGE DISTILLATION

Knowledge distillation [15, 17, 27, 14] transfers knowledge from a large teacher model to a compact student model via softened output distributions, using temperature scaling in softmax to preserve nuanced class relationships. This compression technique enables efficient deployment while maintaining accuracy by mimicking the teacher's decision boundaries and feature representations.

Knowledge distillation methods are categorized into logit-, feature-, and relation-based approaches. Logit-based methods started with compressing ensembles via "soft targets" [15], later including teacher-free online methods [8] and multi-step assistants [32], with recent work aligning predictions at various levels [18, 53]. Feature-based methods transfer internal representations, from layer-wise matching [48] and mutual-information maximization [2] to cross-stage review [10], lightweight projectors [9], and attention-map transfer [13]. Relation-based approaches preserve inter-sample or inter-class relations via distance, correlation, or similarity constraints [35, 37, 41, 16]. Diffusion-model distillation applies conditional consistency for high-fidelity generation [31].

Existing methods typically set important parameters, such as the distillation loss weight, statically during training. However, these parameters often have a significant impact on the training process, and fixing them can hinder the distillation from producing the optimal student model. Our method improves upon this by dynamically adjusting the distillation loss weight during training. Experiments show that this weight affects the quality of the student model, and dynamic adjustment leads to better distillation results.

### 4.2 DYNAMIC WEIGHTING ON KNOWLEDGE DISTILLATION

Dynamic weighting in knowledge distillation (KD) balances cross-entropy (CE) and distillation losses. Instead of fixed weights, recent methods adapt them to reflect varying signal importance. Instance-level strategies [50, 12] use teacher–student discrepancy or sample difficulty, while class- or region-based methods [46, 26] focus on salient regions or distribution similarity. Other works [30, 21, 47, 5, 3, 40, 44] apply confidence-driven reweighting, teacher influence decay, adaptive divergences, or influence-function-based weighting. However, most remain domain-specific and rely on batch-local heuristics, leaving open whether the learned weights enhance CE generalization.

Multi-task learning (MTL) [7, 38, 43, 29, 6, 42, 23, 45, 56, 52, 25, 34] faces a similar challenge of balancing multiple objectives, as joint optimization often leads to gradient conflicts and task interference. Research in this field has followed two major directions: architectural improvements [42, 1, 24], which aim to allocate shared and task-specific capacity more effectively, and optimization strategies, which dynamically adjust task weights or gradients. Representative examples include uncertainty-based weighting [19], gradient-norm balancing [11], and multi-objective optimization formulations such as MGDA [39], along with gradient modification techniques like PCGrad [49], NashMTL [33], and FairGrad [4]. These methods provide more principled and stable optimization compared to heuristic reweighting, but they assume tasks are equally important and aim to improve performance across all tasks, which is misaligned with the asymmetric objectives of KD.

In summary, KD-oriented methods adapt loss weights but lack guarantees for improving CE generalization, while MTL approaches offer principled optimization but fail to reflect KD's asymmetric task roles. Our work addresses this gap by proposing a dynamic weighting framework tailored to KD, explicitly prioritizing CE as the primary objective while leveraging KL as auxiliary guidance.

## 5 CONCLUSION

We addressed the long-standing challenge of balancing CE and KL losses in knowledge distillation. Unlike prior static or batch-local dynamic weighting methods, our framework explicitly optimizes loss weights to promote CE generalization on unseen data, while treating KL as an auxiliary regularizer. By incorporating gradient alignment and normalization, our method ensures stable updates and avoids scale-induced bias. Experiments confirm that this principled strategy yields more consistent knowledge transfer and improved student performance. In the future, we plan to extend this generalization-oriented weighting scheme to broader multi-task and multi-modal learning settings.

# 6 ETHICS STATEMENT

This work adheres to the ICLR Code of Ethics. In this study, no human subjects or animal experimentation was involved. All datasets used were sourced in compliance with relevant usage guidelines, ensuring no violation of privacy. We have taken care to avoid any biases or discriminatory outcomes in our research process. No personally identifiable information was used, and no experiments were conducted that could raise privacy or security concerns. We are committed to maintaining transparency and integrity throughout the research process.

# 7 REPRODUCIBILITY STATEMENT

We have made every effort to ensure that the results presented in this paper are reproducible. The experimental setup, including training steps, model configurations, and hardware details, is described in detail in the paper. Additionally, the datasets used in the paper are publicly available, ensuring consistent and reproducible evaluation results. We believe these measures will enable other researchers to reproduce our work and further advance the field.

# 8 LLM USAGE

Large Language Models (LLMs) were used to aid in the writing and polishing of the manuscript. Specifically, we used an LLM to assist in refining the language, improving readability, and ensuring clarity in various sections of the paper. The model helped with tasks such as sentence rephrasing, grammar checking, and enhancing the overall flow of the text.

It is important to note that the LLM was not involved in the ideation, research methodology, or experimental design. All research concepts, ideas, and analyses were developed and conducted by the authors. The contributions of the LLM were solely focused on improving the linguistic quality of the paper, with no involvement in the scientific content or data analysis.

The authors take full responsibility for the content of the manuscript, including any text generated or polished by the LLM. We have ensured that the LLM-generated text adheres to ethical guidelines and does not contribute to plagiarism or scientific misconduct.

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
