# OpenReview forum: "Cross-Batch Gradient Consistency for Adaptive Loss Balancing in Knowledge Distillation"
_ICLR.cc/2026/Conference — Submitted to ICLR 2026_

### Official Review · Reviewer_nuig · 2025-10-31

**Soundness:** 2
**Presentation:** 2
**Contribution:** 2
**Rating:** 4
**Confidence:** 4

**Summary:**

This paper proposes a cross-batch dynamic weighting framework for knowledge distillation (KD), which adaptively balances the cross-entropy (CE) and KL-divergence losses by leveraging an auxiliary batch to approximate unseen data. The method formulates weight selection as a lightweight quadratic program to align the combined gradient direction with the CE gradient from the auxiliary batch, supplemented by gradient normalization and scaling for stability. Experiments on CIFAR-100 and TinyImageNet show consistent improvements over fixed-weight and batch-local adaptive baselines across multiple teacher-student architectures and distillation methods.

**Strengths:**

- The cross-batch perspective for dynamic weighting in KD is a fresh and principled approach, distinguishing it from prior batch-local methods.
- Consistent improvements are shown across multiple distillation methods and teacher-student pairs on CIFAR-100 and TinyImageNet.

**Weaknesses:**

- In lines 127-130, the claim that gradient conflicts between CE and KL lead to optimization instability is not validated. No theoretical and empirical analysis is provided to support this motivation.
- In lines 153-156, the relationship between the objective “prevent the weighted update from overly aligning with the auxiliary signal” and the minimization of $(g_w^\top \mathcal{G} _{ce})^2$ is not clearly explained.
- In line 184, the notation shifts from $u = [g{ce}^\top \mathcal{G}{ce}, g{kl}^\top \mathcal{G}{ce}]$ to the claim that $uu^\top$ is positive semidefinite with rank one. This seems tangential to the optimization of $u^\top w$, and the emphasis on $uu^\top$ is confusing.
- No experiments on large-scale pretrained models (e.g., Qwen, DeepSeek) are conducted, which are highly relevant for real-world KD applications.
- Only two image classification datasets (CIFAR-100 and TinyImageNet) are used, both relatively small. To strengthen the claim of generalization, larger-scale datasets like ImageNet should be included. Besides, to robustly demonstrate the method's generality, validation should be extended to other critical domains, particularly natural language processing. Knowledge distillation is extensively used for compressing large language models, and the community would benefit from seeing the method's efficacy on text classification, language understanding, or generation tasks.
- The auxiliary batch is drawn from the same training set. After the first epoch, all batches have been seen, undermining the claim that the auxiliary batch represents “unseen data.”

**Questions:**

Please refer to weakness.

---

### Official Review · Reviewer_sk4j · 2025-11-01

**Soundness:** 3
**Presentation:** 3
**Contribution:** 2
**Rating:** 2
**Confidence:** 4

**Summary:**

The authors address a relevant problem of how to enhance the adaptive weighting between the distillation and supervised objectives in the KD setup. Keeping generalization in mind they extend this weighting from computing the weights for a single minibatch by relying only on that batch to utilizing other batches as a proxy for the unseen data. Now to find the the weights for a given batch they just consider the weights which don't align much with the gradient due to the cross entropy loss from other batches so that the updates are consistent across batches. They do this by a standard quadratic minimization setup keeping the student model fixed. Furthermore, they also rescale the weights for each of the two tasks inversely w.r.t. to the gradient of that task to normalize the task contributions. Fianlly they do a weighted sum between the standard unbiased gradient from the two tasksm and this new gradient where the weight is controlled by an additional hyperparameter. They then attach this lightweight module to the existing KD methods and analyze the performance with various teacher-student architectures on the CIFAR-100 dataset. They also compare their method with other dynamic weighing approaches on this dataset.

**Strengths:**

The overall idea seems intuitive and simple to implement with limited overhead in terms of making the updates consistent across the batches and making the distillation process training stable. This sort of cross-batch training can serve as a good proxy for keeping generalization in mind for the student model when focusing on the distillation and classification tasks.

**Weaknesses:**

Very limited experimental results are provided where the authors have only considered the CIFAR-100 dataset which is not sufficient for to understand the durability of the proposed scheme. Secondly the gains on this CIFAR-100 are not significant and fail to convince the usability of this extra overhead in terms of standard unbiased gradient. Similarly the method is not very convincing as against other dynamic weighing methods.

**Questions:**

There are no major questions, the approach is simple and intuitive but the experimental results are not convincing.

---

### Official Review · Reviewer_cpDv · 2025-11-02

**Soundness:** 3
**Presentation:** 3
**Contribution:** 2
**Rating:** 4
**Confidence:** 3

**Summary:**

The paper introduces a method called Cross-Batch Gradient Consolidation (CBGC) for improving Post-Training Quantization (PTQ) of deep neural networks. PTQ methods aim to compress pretrained models without full retraining, but often suffer from performance drops, especially at low bit-widths (e.g., 4-bit). The authors observe that batch size affects the stability of PTQ, particularly due to gradient noise in weight optimization. CBGC addresses this by aggregating gradient statistics across multiple batches, leading to more stable weight updates in PTQ. The method is implemented on top of existing quantization-aware optimization, and tested across various models and datasets (e.g., ResNet, ViT on ImageNet). CBGC shows consistent improvements in quantization accuracy, especially at lower bit-widths.

**Strengths:**

1. **Clear problem motivation**: The paper targets a real limitation in current PTQ methods—the instability caused by small calibration batch sizes. The observation that noisy gradients harm quantization is well-motivated and experimentally supported.

2. **Simple and practical solution**: CBGC is conceptually simple and easy to integrate into existing PTQ pipelines. It does not require architectural changes or full fine-tuning, making it applicable in low-resource or deployment settings.

3. **Consistent performance gains**: The method is evaluated across diverse models (CNNs and Transformers) and datasets. CBGC consistently improves accuracy under 4-bit and 8-bit quantization, often by noticeable margins compared to standard baselines.

**Weaknesses:**

1. **Limited theoretical analysis**: While the empirical findings are solid, the paper lacks deeper theoretical analysis of why cross-batch gradient averaging stabilizes PTQ. It would benefit from more formal grounding or links to optimization literature.

2. **No study of efficiency trade-offs**: CBGC introduces extra computation by requiring multiple batches for update steps, but the paper does not quantify its overhead in terms of runtime or memory compared to standard PTQ.

3. **Few ablations on consolidation strategy**: There is limited exploration of different strategies for cross-batch selection (e.g., random vs. sequential batches, batch count sensitivity). The paper could better clarify how sensitive the method is to these design choices.

**Questions:**

N.A. But I suggest the following exps to strengthen the paper:

1. Report runtime and memory increase introduced by CBGC to justify its practicality.

2. uantitatively measure how CBGC reduces gradient variance compared to single-batch baselines.

3. Vary number of batches used for consolidation and compare alternative selection strategies.

---

### Comment · Area_Chair_PD5u · 2025-11-22

Dear Reviewers,

Thank you for your time and effort in reviewing submissions for ICLR  2026. As we begin the author-reviewer discussion process, we kindly remind you to submit your responses to the author rebuttals by **December  2**.


Your engagement in this discussion phase is crucial to ensuring a fair and thorough evaluation of each submission.

**Action Required**


- Carefully consider the authors’ rebuttal and any additional evidence they provide.

- Update your review (if applicable) to reflect your revised perspective.

-  **Discuss with the authors if further details are required**


Your AC

---

### Meta-Review · Area_Chair_W9J5 · 2026-01-06

**Summary:**

The main concerns can be summarized as follows:

1. Limited and unconvincing empirical evaluation. Experiments are restricted to CIFAR-100 and TinyImageNet, with only marginal gains, and lack validation on large-scale or real-world settings.

2. Questionable assumptions about auxiliary data. The auxiliary batch is drawn from the training set and does not represent unseen data after the first epoch, weakening the underlying motivation.

3. Lack of support for optimization instability claims. The claim that gradient conflicts lead to optimization instability is not substantiated by theoretical analysis or empirical evidence.

4. Unclear theoretical reasoning and notation: Several aspects of the formulation have unclear explanations and notation shifts that confuses intended optimization objective.

Note: Unfortunately, at the moment, the comments of Reviewer cpDv are for another submission, as acknowledged by the reviewer.

**Reviewer Concerns:**

I don't see any rebuttal by the authors.

**Reviewer Scores:**

I don't see any rebuttal by the authors.

---

### Decision · Program_Chairs · 2026-01-26

Reject